# Critical Role of Iodine and Thyroid Hormones During Pregnancy

**DOI:** 10.3390/ijms262110247

**Published:** 2025-10-22

**Authors:** Rodrigo Moreno-Reyes, Camilo Fuentes Peña, Jonathan F. Nuñez, María Belén Sánchez, Jonatan J. Carvajal, Katherine Roble, María José Mendoza-León, Ma. Andreina Rangel-Ramírez, Ma. Cecilia Opazo, Margarita K. Lay, Claudia A. Riedel, Enrique Guzmán-Gutierrez, Juan Pablo Mackern-Oberti, Evelyn L. Jara

**Affiliations:** 1Department of Nuclear Medicine, Hôpital Universitaire de Bruxelles, Hôpital Erasme, Université Libre de Bruxelles, Route de Lennik 808, 1070 Brussels, Belgium; rodrigo.moreno-reyes@ulb.be (R.M.-R.); camilo.fuentes.pena@ulb.be (C.F.P.); 2École de Santé Publique, Université Libre de Bruxelles, 1070 Brussels, Belgium; 3Departamento de Farmacología, Facultad de Ciencias Biológicas, Universidad de Concepción, Concepción 4070386, Chile; jonathan.fernando.n@gmail.com; 4Instituto de Medicina y Biología Experimental de Cuyo, IMBECU CCT Mendoza—CONICET, Mendoza CP5500, Argentina; bsanchez@mendoza-conicet.gob.ar (M.B.S.); jpmackern@mendoza-conicet.gob.ar (J.P.M.-O.); 5Facultad de Veterinaria, Universidad Juan Agustín Maza, Mendoza CP5519, Argentina; 6Departamento de Biotecnología, Facultad de Ciencias del Mar y Recursos Biológicos, Universidad de Antofagasta, Antofagasta 1270300, Chile; jonatan.carvajal.c@gmail.com (J.J.C.); margarita.lay@uantof.cl (M.K.L.); 7Departamento de Bioquímica Clínica e Inmunología, Facultad de Farmacia, Universidad de Concepción, Concepción 4070386, Chile; kroble2017@udec.cl (K.R.); eguzman@udec.cl (E.G.-G.); 8Millennium Institute on Immunology and Immunotherapy, Santiago 8320000, Chile; mariaj.mendoza12@gmail.com (M.J.M.-L.); rangelmariandre@gmail.com (M.A.R.-R.); claudia.riedel@unab.cl (C.A.R.); 9Centro de Investigación de Resiliencia a Pandemias, Facultad de Ciencias de la Vida, Universidad Andrés Bello, Santiago 8320000, Chile; 10Instituto de Ciencias Naturales, Facultad de Medicina Veterinaria y Agronomía, Universidad de Las Américas, Santiago 7500975, Chile; mopazod@udla.cl; 11Centro de Investigación en Ciencias Biológicas y Químicas, Universidad de Las Américas, Santiago 7500975, Chile; 12Research Center in Immunology and Biomedical Biotechnology of Antofagasta (CIIBBA), University of Antofagasta, Antofagasta 1270300, Chile; 13Instituto de Fisiología, Facultad de Ciencias Médicas, Universidad Nacional de Cuyo, Mendoza CP5500, Argentina

**Keywords:** thyroid hormones, iodine, pregnancy, placenta, hypothyroidism

## Abstract

Iodine is an essential micronutrient that is required for thyroid hormone (TH) synthesis. However, adequate maternal thyroid function is critical for fetal growth and neurodevelopment. Pregnancy increases iodine requirements due to enhanced renal clearance, higher maternal TH production, and transplacental transfer, making pregnant women especially vulnerable to iodine deficiency. In this review, we examine the molecular mechanisms of TH synthesis and regulation, placental transport and metabolism, and the physiological adaptations of thyroid function during gestation. We also analyze the clinical and public health consequences of iodine imbalances, ranging from deficiency to excess. Evidence indicates that mild iodine deficiency—which is common even in developed countries—can lead to maternal thyroid overstimulation, increased thyroglobulin levels, altered T3/T4 ratios, and enlarged thyroid volume, while severe deficiency results in maternal and fetal hypothyroidism with irreversible neurocognitive impairment in the offspring. Conversely, excessive iodine intake may impair fetal thyroid function through mechanisms such as the Wolff–Chaikoff effect. In conclusion, ensuring balanced iodine intake through iodized salt, supplementation, and routine thyroid monitoring during pregnancy is essential to protect maternal health and optimize early neurodevelopment.

## 1. Introduction

Iodine is an essential trace element that is required in small amounts for the synthesis of thyroid hormones (THs), including tri-iodothyronine (T3) and tetraiodothyronine or thyroxine (T4) [1]. THs are crucial regulators of many physiological processes, especially during pregnancy and fetal development, and require iodine as a key substrate for their synthesis. Adequate iodine intake during pregnancy is therefore vital, as it directly affects maternal thyroid function and fetal growth [2]. The physiological changes that occur during pregnancy, including increased TH production, renal iodine clearance, and fetal TH synthesis, increase the need for iodine [3]. As a result, pregnant women are especially susceptible to iodine deficiency, which can negatively impact both maternal health and fetal development, especially in terms of the neurological and cognitive development of the fetus [4,5,6].

Iodine deficiency in pregnancy is a significant global public health issue, affecting millions of women and their developing offspring. The World Health Organization (WHO) recommends a daily iodine intake of 250 μg during pregnancy, compared to 150 μg for non-pregnant women [7,8]. Despite this recommendation, achieving sufficient iodine intake remains a challenge in many regions, even in developed countries. Low iodine levels during pregnancy have been linked to harmful and persistent effects such as impaired fetal neurodevelopment and cognitive function, with evidence showing that children born to iodine-deficient mothers exhibit increased risks of cognitive impairments, including decreased intelligence quotient (IQ) [4,5,6]. The relationship between maternal thyroid function and fetal development is complex and multifactorial. The fetus relies entirely on maternal THs during early pregnancy, as its own thyroid gland only becomes functional toward the end of the first trimester [3]. As this critical period includes important developmental stages such as neurogenesis, neuronal migration, and early myelination, maintaining good maternal thyroid health through sufficient iodine intake is vital for healthy fetal development [9]. This review aims to provide a comprehensive analysis of the mechanisms regulating TH homeostasis and to explore the effects of iodine nutrition during pregnancy.

## 2. Thyroid Hormones in Pregnancy

THs are essential for maternal health and fetal development, especially in early pregnancy when the fetus depends entirely on maternal THs. Pregnancy causes physiological changes such as increased levels of human chorionic gonadotropin (hCG), higher estrogen, and increased renal iodine clearance, which influence maternal thyroid function and iodine requirements [3,10]. While the sufficient production of T4 and T3 supports both placental development and proper fetal neurodevelopment [11,12], maternal iodine deficiency can result in negative outcomes, including miscarriage, preterm birth, poor fetal growth, and long-term neurodevelopmental issues [13]. Therefore, ensuring adequate iodine intake for TH synthesis and monitoring thyroid health are essential during both prenatal and pregnancy stages.

### 2.1. Thyroid Hormones Synthesis

THs are synthesized and released by the thyroid gland [14,15]. Anatomically, the thyroid gland is one of the largest endocrine glands in the neck [16], consisting of two lobes connected by a structure called the isthmus [17]. The hormones produced by the thyroid include 3,5,3′,5′-tetraiodothyronine or thyroxine (T4) and 3,5,3′-triiodothyronine (T3) [18]. TH biosynthesis requires iodine (I), which is an essential component of both T3 and T4 and is supplied through the diet as iodide (I^−^) [19]. Iodide is absorbed in the small intestine, facilitated by the sodium/iodide symporter (NIS) on the apical membrane of intestinal cells [20]. Subsequently, iodide is secreted into the stomach and salivary gland lumens via the NIS on the basolateral membrane. This iodide is reabsorbed into the bloodstream via the NIS on the apical membrane in the small intestine, completing its cycling back into systemic circulation [21]. Circulating iodide is either absorbed by the thyroid gland through the NIS on the basolateral plasma membrane of thyrocytes or excreted from the body through the urine [1].

The synthesis and release of THs are initially stimulated by the production of thyrotropin-releasing hormone (TRH) in the hypothalamus, which stimulates the synthesis and secretion of thyroid-stimulating hormone (TSH) in the anterior pituitary [22,23]. Next, TSH binds to its receptor on the thyroid gland, stimulating the biosynthesis and secretion of THs [23]. This regulatory pathway is part of the neuroendocrine system known as the hypothalamic–pituitary–thyroid (HPT) axis [24]. Additionally, THs regulate the secretion of TRH and TSH through a negative feedback loop [22]; in particular, THs inhibit the transcription of TRH and TSH subunit genes, as well as the post-translational modification and release of TSH [23].

The biosynthesis of THs occurs within the thyroid follicles, which are the functional units of the thyroid consisting of thyrocytes that form well-defined anatomical structures. The apical side of thyrocytes is the lumen, where synthesis and storage take place, while the basolateral side contacts with the blood capillaries, facilitating the release of hormones into circulation [17,25]. Iodide (I^−^) is transported from the bloodstream into the thyrocytes via NIS [26]. Subsequently, efflux from the apical membrane toward the lumen is mediated by ionic channels such as pendrins [27] and anoctamin [28]. Thyrocytes also produce and secrete thyroglobulin (TG) into the lumen [29]. Then, thyroperoxidase (TPO) catalyzes the oxidation of I-, enabling its incorporation into tyrosine residues of TG to produce iodinated TG [25]. This enzymatic oxidation occurs in the presence of hydrogen peroxide (H_2_O_2_) [30], produced by the thyroid NADPH oxidase called dual oxidase 2 (DUOX2) [31,32]. Iodinated TG forms 3-iodotyrosine or monoiodotyrosine (MIT) and 3,5-diiodotyrosine or diiodotyrosine (DIT) [33], following which T3 and T4 are formed through the coupling of MIT and DIT [34]. During this reaction, the oxidation of iodotyrosines is also catalyzed by TPO [18]. Finally, thyrocytes endocytose TG from the follicular lumen via endocytic vesicles. In this process, THs are proteolyzed from iodinated TG to ultimately release intrathyroidal iodide, with this step mediated by iodotyrosine dehalogenase (DEHAL1) [35]. Subsequently, THs are released into the bloodstream at the basolateral membrane through monocarboxylate transporters (MCTs) 8 and 10 [36]. The mechanism of TH biosynthesis can be summarized in four stages: oxidation of TPO by H_2_O_2_, iodination reaction, coupling reaction, and iodide recycling along with the release of THs from follicles [25] (Figure 1). T4, often considered a prohormone or reserve hormone, is the most abundantly secreted hormone by the gland [37].

Serum transport of T4 occurs through its strong binding to the main plasma thyroid hormone-binding proteins, including thyroxine-binding globulin (TBG), transthyretin (TTR), and albumin [38]. Although TBG has the highest affinity for THs in humans, TTR also significantly contributes to TH transport in specific tissues such as the placenta [39,40]. These proteins change under various conditions, such as different gestational stages and hyperthyroidism [41]. Deiodination also takes place in peripheral tissues, activating THs and regulating their biological functions. This process is mediated by tissue enzymes with different expression patterns, enabling dynamic control of TH signaling [42]. The enzymes which catalyze deiodination are known as type 1 (D1), type 2 (D2), and type 3 (D3) deiodinases [43]. T3—considered the biologically active hormone—is produced by deiodination of T4 in peripheral tissues, mainly by D1 and D2 [32,33]. Conversely, D3 inactivates T4 by converting it to reverse T3 (rT3) [43]. Therefore, it is proposed that the activation of deiodinases D1 and D2, along with the inactivating enzyme D3, can modulate TH activities independently of changes in serum TH concentrations, depending on the requirements of specific tissues [42,44].

Regarding the mechanisms of action of THs, two effects are known: the genomic effect, which involves the direct action of THs on gene transcription, and non-genomic effects. The genomic effect begins with the entry of thyroid hormones into the cell, where T4 is converted into T3 by D2. Once converted, T3 binds to the nuclear receptors (TRs) of THs, which then attach to the promoter regions of target genes [45,46,47,48,49,50]; for example, TRs may bind to genes necessary for producing myelin basic protein in the central nervous system (CNS) [51], or to genes such as Endothelin-1 and fibronectin in umbilical cord vein endothelial cells (HUVECs) [49]. On the other hand, non-genomic effects start with THs binding to their membrane receptor, integrin αvβ3. This activates a signaling cascade that leads to cell proliferation; angiogenesis; secretion of growth factors in the membranes of erythrocytes, leukocytes, or hepatocytes; intracellular trafficking of proteins; and other functions [52,53,54]. It has been shown that both T3 and T4 can bind to the αvβ3 receptor; however, they bind to different domains and activate distinct signaling pathways [55]. T3 binds to the S1 domain, which activates phosphatidylinositol 3 kinase (PI3K) and promotes, via TRα, the transcription of hypoxia-inducible factor-1α (HIF-1α). Meanwhile, both T3 and T4 bind to the S2 domain, activating extracellular signal-regulated kinase (ERK1/2) which, through TRβ1, promotes cell proliferation [56]. Additionally, the receptor’s own αv subunit responds to T4 by entering the nucleus and binding to the promoter regions of the HIF-1α, TR-β1, and cyclooxygenase-2 genes, thereby promoting their transcription [54].

### 2.2. Thyroid Hormones and Placenta

The placenta is a transient, highly specialized organ that facilitates gas and nutrient exchange while serving as a selective barrier. It also plays a crucial role in regulating the concentrations of THs in the maternal and fetal circulations, ensuring proper fetal development [40]. As the fetus initially relies entirely on maternal THs, any disruption in their production or regulation can lead to complications such as pre-eclampsia or miscarriage [57]. The biological activities and cellular effects of THs depend on their interactions with membrane transporters, specific receptor proteins, and their intracellular metabolism, which is primarily mediated by iodothyronine deiodinases (D) [58]. During pregnancy, THs cross the placenta via thyroid hormone transporters (THTs), including monocarboxylate transporters (MCT8 and MCT10), L-type amino acid transporters (LAT1 and LAT2), and organic anion transporters (OATP1A2 and OATP4A1) [40]. These transporters are expressed differently in various placental cell types [59]. For example, MCTs, OATPs, and LAT1 are found in villous syncytiotrophoblasts; MCTs and OATP1A2 are present in cytotrophoblasts and extravillous trophoblasts; and MCTs are additionally expressed in the stroma [46]. THRs are widely expressed in placental structures, with THRα1 being the predominant isoform in the human term placenta. Strong THR expression has been detected in syncytiotrophoblasts, villous and extravillous cytotrophoblasts, and chorionic villi [60]. In the decidua, stromal and endothelial cells express TRα1, TRα2, and TRβ1, with their expression patterns changing throughout gestation [60,61]. D3 is the most highly expressed deiodinase in the placenta [62]. Placental D expression varies throughout pregnancy: D2 is more abundant in early gestation but declines toward term, while D3 expression increases. Immunohistochemical studies have shown predominant D3 expression in syncytiotrophoblasts during the first trimester, with D2 becoming more prominent later in pregnancy [63]. These findings suggest that D3 may play a key role in regulating maternal T3 transfer to the fetus. Additionally, T4 can be transferred from the maternal to fetal circulation through the chaperone action of TTR, which facilitates TH transport across the placental barrier to the fetal capillaries [64].

## 3. Iodine in Pregnancy

The interaction between iodine metabolism and fetal development is a complex physiological process that changes throughout pregnancy, with requirements that differ based on developmental stages and maternal adaptations. Understanding these dynamic needs requires the examination of three interconnected factors: the regulatory roles of THs in development, iodine metabolism pathways during gestation, and specific quantitative requirements for optimal outcomes.

### 3.1. The Critical Role of Thyroid Hormones for Fetal Development

Fetal development is complex and highly regulated by different factors. The process begins with gastrulation and the formation of three germ layers known as the ectoderm, mesoderm, and endoderm [65]. The thyroid gland develops from the endoderm after 4 weeks of gestation and is located at the base of the primitive pharynx. At this site, a thickening called the thyroid diverticulum forms. Subsequently, the thyroid diverticulum leads to the development of thyroid follicles by around 8 weeks of gestation. Finally, once the thyroid follicles mature (at approximately 6 weeks into gestation), they form the thyroid gland. It is worth noting that the thyroid gland also contains parafollicular cells (also known as C cells), which have neuroendocrine properties and mainly produce calcitonin—a hormone involved in calcium and phosphate regulation [66]. Meanwhile, the CNS develops from the ectoderm, beginning about 20 days after gastrulation. Specifically, the ectoderm thickens in the dorsal region of the embryo to form the neural plate, which then gives rise to the neural folds. These neural structures eventually shape the neural tube, which develops into the CNS [67]. The fetal brain is highly plastic and vulnerable to environmental and nutritional influences, and alterations in these factors can impact fetal health and development.

One key factor in fetal development and maturation is the thyroid gland, which is responsible for secreting and producing the body’s THs, including T3 and thyroxine T4. These hormones play vital roles in overall growth, energy regulation, and neurodevelopment within vertebrates [68]. Thyroid cells do not begin secreting T3 and T4 until after 12 weeks of gestation, with significant concentrations observed at around 18 to 20 weeks [69]. After birth, T3 and T4 concentrations increase rapidly [70]. The secretion of THs is tightly controlled, primarily by TSH secreted by the pituitary gland. Premature babies have been reported to have lower TH concentrations, potentially due to issues within the HPT axis. Additionally, infants born to mothers with severe autoimmune hypothyroidism tend to have lower TH concentrations, caused by the transfer of maternal anti-thyroid antibodies through the placenta [71,72,73,74].

As mentioned earlier, THs are essential for numerous processes, especially in fetal neurological development; in particular, they are crucial for neuronal migration, differentiation, myelination, and cell signaling [9]. A TH deficiency during early gestation can cause significant physiological and morphological developmental issues, as these hormones are fundamental for the differentiation and maturation of critical brain regions such as the cerebral cortex, hippocampus, and cerebellum [74,75]. Both congenital and maternal hypothyroidism can lead to cretinism, characterized by intellectual impairment, deaf-muteness, gait disturbances, and various cognitive problems. Studies have shown that low TH concentrations can affect the gray matter volume [75]. Moreover, maternal THs transferred during gestation (particularly during the first trimester) are involved in brain maturation, neuron proliferation and migration, neuronal and glial cell differentiation, and synaptogenesis [72]. Changes in TH concentrations can also alter brain chemistry. A neuronal population that is sensitive to TH deficiency is the γ-aminobutyric acid (GABA)-ergic interneurons in the neocortex, hippocampus, and cerebellum [76], with decreased GABA reducing cell migration and axon myelination in the CNS [77]. Other abnormalities linked to low maternal THs include limited axon and dendrite growth, reduced neuronal connectivity, myelin deficits, and decreased synaptic densities [78]. Conversely, elevated maternal TH concentrations accelerate neurological development, leading to increased cell proliferation and differentiation but potentially resulting in decreased brain and body tissue weight and brain degeneration [79]. Experimental hyperthyroidism in pregnant Wistar rats caused morphological alterations in the placenta, such as a thinner basal zone and increased decidua thickness, along with higher fetal, placental, and basal zone weights on day 19 of gestation [80]. These findings suggest that hyperthyroidism affects both placental function and fetal growth, which may be related to the preterm labor observed in pregnant women with hyperthyroidism [81].

### 3.2. Iodine Metabolism During Pregnancy

Dietary iodide and recycled iodide from thyroid hormone (TH) metabolism are absorbed in the small intestine, aided by expression of the sodium-iodide symporter (NIS) on the apical side of enterocytes [20], allowing iodide to enter the bloodstream. While the recommended dietary iodine intake for adults is 150 μg per day, the need for iodine increases during pregnancy due to higher production of THs, increased iodide clearance and transfer to the fetus, and the increase in type III deiodinase activity [20]. When the iodide supply is insufficient, the thyroid gland adapts by increasing iodide uptake [82]. Deiodination is the first step in the activation or inactivation of THs, involving the removal of one iodine atom from the outer tyrosyl ring of T4 to produce T3, which is considered the biologically active form. D1 and D2 catalyze the outer ring deiodination of T4 and convert it to the active hormone T3 [82]. D3 converts T4 to reverse T3 and T3 to T2, and is highly expressed in the pregnant uterus, placenta, and fetal and neonatal tissues [82]. Finally, T4 interacts with plasma membrane receptors—specifically, integrin ανβ3—and triggers non-genomic effects such as promoting angiogenesis and tumor cell proliferation. Additionally, the binding of T4 to integrin ανβ3 influences the activity of specific membrane ion pumps and stimulates protein trafficking within the cell [54,56,83].

### 3.3. Iodide Requirement During Pregnancy

The increase in iodine requirements during pregnancy due to urinary losses [84,85] results in increased TH production, and the transfer of iodine to the fetus is compensated for by increased thyroid iodide uptake [86,87,88,89,90]. Therefore, although the recommended adequate iodine intake for adults is 150 μg/day, the WHO recommends an iodine intake of 250 μg/day for pregnant women (Figure 2D) [7]. Urinary iodine concentration (UIC) is a sensitive indicator of current iodine intake, as roughly 90% of all consumed iodine is excreted in the urine [91]. The median UIC indicative of optimal iodine intake in pregnant women ranges between 150 and 249 µg/L [92]. In pregnant women with adequate iodine levels, the increased iodide requirements due to renal losses and fetal iodide transfer often go unnoticed (Figure 2D); however, for those with borderline low iodine intake, pregnancy can exacerbate iodine deficiency, stressing the homeostatic mechanisms that maintain a euthyroid state [6]. A healthy, non-pregnant woman typically needs about 80 µg of iodide daily to produce THs. When iodide intake is optimal at 150 µg/day, a thyroid uptake of 35% is sufficient to maintain balance and support iodine stores, which range from 10 to 20 mg (Figure 2A). Of the 80 µg of iodide ingested, 15 µg is lost through the feces, while the remaining 65 µg is divided between the thyroid and renal losses [93]. When iodide intake is low, the thyroid compensates by increasing its iodide uptake to meet the daily requirement of 80 µg for production of THs. Despite this, a daily deficit of around 12 µg occurs, leading to a gradual depletion of thyroid iodine reserves (Figure 2B). In iodine-deficient pregnant women, the iodide requirement increases from 80 to 120 µg/day due to a 50% increase in TH production (Figure 2C). Although the thyroid increases iodide uptake, the abovementioned total iodide intake is insufficient to meet the heightened requirement for TH production and the daily iodide deficit rises to about 20 µg/d [6,94].

## 4. Physiological Adaptation of Thyroid Function During Pregnancy

The risk of iodine deficiency appears to be higher in women than men [95], making women prone to iodine deficiency. A combination of dietary, physiological, and lifestyle factors may explain why women tend to have lower iodine intake than men (Figure 3). Dietary preferences are one factor, as women are more likely to follow plant-based diets which are low in iodine. Physiological factors such as pregnancy and menstrual blood loss may also contribute to iodine deficiency. Pregnancy can increase the risk through a dual mechanism: it raises iodine requirements, and multiple pregnancies can cumulatively deplete iodine stores. Additionally, pregnancy and menstrual blood loss are often linked with iron deficiency. Iron deficiency and iron-deficiency anemia are widespread in women and may affect thyroid function [96,97,98]; in particular, iron deficiency decreases the iodination activity of TPO—an enzyme containing a heme group—which consequently impairs iodine organification and uptake. The effects of combined iodine and iron deficiency may also help to explain the higher risk of iodine deficiency and prevalence of thyroid diseases in women compared to men [99].

Iodine status is typically assessed via a population’s UIC; however, this may not accurately reflect the status of an individual or subgroup of the population. The drawback of using UIC for individual assessment is that it can vary significantly from day to day due to factors such as hydration, which influences urinary dilution [100].

Iodine requirements are increased during pregnancy, due to the heightened production of THs and transfer of iodine to the fetus, in addition to increased renal iodine clearance [10]. While iodine-sufficient pregnant women can adapt to these increased iodine requirements without visible effects on the thyroid gland, the thyroid’s ability to maintain optimal function may be impaired in cases of iodine deficiency, leading to functional and structural changes in the maternal and fetal thyroid glands.

Pregnancy induces several physiological modifications of thyroid function, caused by increased TBG concentrations due to estrogen stimulation of hepatocyte protein synthesis by week 7 of gestation. Compared to pre-conception levels, TBG concentrations are 2.5 times higher than in non-pregnant women [101,102]. This increase in serum TBG leads to higher total T4 and T3 levels, associated with a slight decrease in free THs [95,103]. T4 synthesis increases during pregnancy to compensate for the increased hormone-binding capacity of TBG [15], helping to maintain optimal free hormone concentrations [104]. By the end of the third trimester, the peak hCG level causes a transient increase in free T4 and a transient decrease in TSH. Under normal conditions, TSH concentrations in the second half of pregnancy are similar to those in non-pregnant women. Conversely, TSH concentrations may increase above pre-conception reference ranges in women with low iodine intake (Figure 4) [2].

## 5. Health Implications of Deficiency and Iodine Excess

The iodine status during pregnancy, from severe deficiency to excess, presents different challenges in terms of both maternal and fetal health. Each level of iodine imbalance manifests through specific clinical and biological signs, with effects that vary in their severity and long-term outcomes. While severe iodine deficiency causes serious developmental issues, mild deficiency can have more subtle effects that may go unnoticed. On the other hand, excess iodine intake introduces its own risks, especially for fetal thyroid health. Recognizing these different signs and their underlying causes is essential for informing effective public health strategies.

### 5.1. Clinical and Biological Consequences of Mild Iodine Deficiency in Pregnant Women and Newborns

Mild iodine deficiency (50–99 µg/L) frequently occurs in pregnant women in many countries, including those in the European region [7]. Mild iodine deficiency can lead to goiter formation and increased thyroid volume (TV) in the newborn. However, TH concentrations tend to remain within normal ranges in pregnant women and newborns [88,105] (Figure 5).

Unlike other nutritional deficiencies where a single blood test can determine deficiency, no individual laboratory test can diagnose iodine deficiency. The absence of a specific marker for iodine status complicates the diagnosis in a given subject, as well as the inclusion of truly iodine-deficient subjects in clinical trials. Most clinical trials use the median UIC of pregnant women in a specific region, assuming that it reflects each woman’s iodine status. It is probable that as many as 90% of pregnant women in mildly iodine-deficient areas, based on the median UIC, show no biological signs of deficiency [105]. This uncertain iodine status among pregnant women introduces bias in the design and interpretation of iodine supplementation trials, especially regarding mild deficiency [84,85]. Although no single parameter can definitively indicate iodine deficiency, a combination of thyroid measurements may help to detect excessive thyroid stimulation in otherwise euthyroid pregnant women. These include low free T4, increased T3/T4 ratio, and elevated TG concentrations. Free T4 concentrations slightly decrease by the end of pregnancy, even with adequate iodine intake. While this decrease is more marked in iodine-deficient women, the values generally remain within normal ranges. In cases of mild deficiency, up to one-third of pregnant women may have free T4 concentrations near or below normal limits [86,104]. Interestingly, newborns often have higher free T4 concentrations than their mothers, indicating fetal protection against hypothyroxinemia [88], and early iodine supplementation has been shown to prevent hypothyroxinemia [86,87,106]. Low iodine intake favors T3 secretion, leading to an elevated T3/T4 ratio [82], while iodine supplementation can mitigate this increase [105]. Under normal iodine conditions, the T3/T4 ratio ranges from 10 to 22 × 10^−3^ in pregnant women [107,108]. After delivery, this ratio may take months to return to normal in women without supplementation, suggesting long-lasting effects of thyroid stimulation [109]. Pregnancy—especially with marginal iodine intake—acts as a goitrogenic factor with potentially accumulating effects, as evidenced by the link between parity and increased TV [89,90,110]. TG concentrations correlate with TV and tend to increase in iodine-deficient areas [2,33,111,112]. TG appears to be more sensitive to iodine status variations than TSH, often being the only parameter associated with UIC in mildly deficient pregnant women [2]. Thus, TSH may be less useful for the early detection of mild iodine deficiency. Elevated TG and diffuse TV (>18 mL) in euthyroid pregnant women can also indicate iodine deficiency [105]. Consequently, routine laboratory tests and standardized criteria can help to identify iodine-deficient pregnant women through the detection of excessive thyroid stimulation. Practically, a euthyroid pregnant woman with a free T4 at or below the lower reference for iodine-sufficient pregnant women—namely, a T3/T4 ratio above 25 × 10^−3^—and a TG at or above the high reference point may be at higher risk of iodine deficiency (Figure 6). These surrogate markers could guide targeted iodine supplementation for those who are most likely to benefit. Nevertheless, it should be noted that current guidelines recommend iodine supplementation for all women planning pregnancy or who are already pregnant. In particular, the WHO advises supplementation in iodine-deficient areas or in countries where iodine prophylaxis programs are not yet consolidated, while the ATA guidelines recommend a daily oral supplement containing 150 μg of iodine regardless of the woman’s iodine nutritional status [7,113].

In summary, the thyroid gland adapts to mild iodine deficiency by increasing thyroid function to maintain the euthyroid state during pregnancy. The first step of this homeostatic mechanism is enhanced iodine uptake triggered by TSH, which occurs due to a slight decrease in free T4. Although this compensatory mechanism achieves its objective, there is a price to pay: chronic hyperstimulation, leading not only to increased TV in pregnant women and newborns, but also to a higher risk of thyroid nodules later in life. Meanwhile, iodine uptake and TH synthesis by the fetal thyroid begin at 10–12 weeks of gestation, and thyroid secretion becomes effective by mid-gestation [114]. The fetal thyroid function relies entirely on the iodine transferred from the pregnant women [114,115]. In cases of mild iodine deficiency, despite the relative hypothyroxinemia observed in pregnant women, the newborn is protected against hypothyroxinemia [88]. Similarly, iodine deficiency stimulates fetal thyroid activity, increasing TG concentrations in the cord blood and TV when compared to iodine-sufficient newborns [2]. Two randomized iodine supplementation trials involving pregnant women in Belgium and Denmark with mild iodine deficiency reported similar findings for cord TSH and TG concentrations [2,86]. Cord TSH concentrations did not differ between iodine-supplemented and non-supplemented pregnant women. In contrast, iodine supplementation led to significant decreases in cord TG concentrations when compared to newborns from non-iodine-supplemented pregnant women in both trials. These results suggest that cord TG may be more sensitive to variations in iodine intake than TSH, as previously shown in an iodine-deficient region of Italy. Additionally, the TV in newborns from the Brussels trial was significantly higher in those born to non-supplemented pregnant women than those born to supplemented pregnant women.

### 5.2. Clinical and Biological Consequences of Severe Iodine Deficiency in Pregnant Women and Newborns

The most severe effects of low iodine intake occur when the iodine deficiency is severe. When the median UIC of school-age children is below 20 μg/L, a population is considered severely iodine deficient [7]. In severe iodine-deficient populations, goiter, cretinism, and hypothyroidism are endemic [116]. Hypothyroidism during pregnancy can lead to both maternal and fetal hypothyroidism, causing severe neurological and cognitive impairments in the offspring. Unlike mild iodine deficiency, the thyroid gland can no longer compensate to maintain normal thyroid function in cases of severe deficiency, and hypothyroidism may develop during or even before pregnancy. A double-blind controlled trial conducted between 1966 and 1982 in an iodine-deficient region of the Western Province of Papua New Guinea evaluated the effectiveness of intramuscular iodized oil as a prophylactic measure against endemic cretinism, compared to saline controls. The study showed that pre-conception iodized oil supplementation prevented endemic cretinism. Additionally, the overall survival rate at 15 years was higher in the treated group than in the control group. Children whose mothers received supplementation demonstrated significantly improved intellectual and motor skills [117,118,119]. A recent 2024 study in Greece investigated how iodine intake affects motor and cognitive development in children aged 4 and 6 years. The results indicated that low iodine intake during childhood (UIC 100 μg/L) was associated with motor impairments in 4-year-olds and reduced IQ in 6-year-olds when compared to a control group [118]. These studies confirm that iodine deficiency during pregnancy causes severe cognitive and motor deficits, while iodine supplementation effectively prevents these deficits.

In some iodine-deficient regions, goitrogens in staple foods may further aggravate iodine deficiency during pregnancy by inhibiting iodine uptake [120,121]. Thiocyanate, which is present in cassava and tobacco smoke, competitively inhibits iodine uptake. Flavonoids in millet impair iodide organification by inhibiting TPO iodination activity [122]. Furthermore, other trace elements such as selenium and iron may influence thyroid metabolism, affecting how the thyroid adapts to iodine deficiency [94,98,123,124]. Endocrine Disruptors (EDCs) are chemicals from outside the body that interfere with the production, processing, or action of hormones, increasing the risk of health problems [125]. Several EDCs, including perchlorate, nitrate, and thiocyanate, block the uptake of iodine by the thyroid [126], and may also disrupt thyroid function during pregnancy [127,128,129]. A study, performed as part of the Controlled Antenatal Thyroid Study (CATS) involving over 21,000 pregnant women in the United Kingdom and Italy, found that high perchlorate exposure in pregnant women was associated with low IQ scores in their 3-year-old children. The researchers found that pregnant women with the highest urine perchlorate concentrations were most likely to have children with the lowest IQ. This finding remained true even after adjusting for hypothyroidism (if present) starting in the first trimester, suggesting that perchlorate may have adverse effects independent of TH status. Overall, pregnant women in general had low urinary iodine, indicating inadequate iodine intake [130]. Meanwhile, a Chinese birth cohort study explored how maternal exposure to perchlorate, thiocyanate, and nitrate affects offspring neurodevelopment, which revealed that a doubling of thiocyanate and nitrate in urine during the first trimester was linked to decreases of 1.56 (95% CI: −2.82, −0.30) and 1.22 (−2.40, −0.03) points, respectively, in the offspring’s mental development index [131]. These findings suggest that prenatal exposure to these chemicals—particularly thiocyanate and nitrate at current levels—may impair neurodevelopment.

### 5.3. Iodine Excess

Acute high iodine intake can cause transient inhibition of TH synthesis through a mechanism known as the acute Wolff–Chaikoff effect [132]. This inhibition is temporary, and the thyroid typically “escapes” the Wolff–Chaikoff effect within a few days by downregulating its iodide uptake, allowing normal TH production to resume [86]. If the escape mechanism fails, it can result in iodine-induced hypothyroidism [133]. The fetal thyroid gland is not fully capable of escaping from the Wolff–Chaikoff effect until late in gestation and, as such, fetal hypothyroidism may occur in cases of acute iodine excess, even if maternal thyroid function remains normal during pregnancy [134].

The Wolff–Chaikoff effect generally occurs with acute iodine excess during medical or diagnostic procedures, which are relatively rare during pregnancy. However, the use of iodine-containing antiseptics—especially during cesarean sections—has been linked to higher neonatal TSH levels, increasing the likelihood of recall for congenital hypothyroidism [135].

Excess iodine from dietary or environmental sources is a more common issue and has been associated with a higher prevalence of thyroid autoimmunity [86,136]. In countries with iodine excess such as Chile, the prevalence of hypothyroidism among the adult population—including pregnant women—is highly elevated [137]. A cross-sectional study in China reported a U-shaped relationship between thyroid function and UIC among pregnant women in their first trimester; furthermore, women with UICs of 150–249 µg/L exhibited the lowest serum TSH and TG concentrations [138]. The WHO-recommended safe upper limit for iodine intake in pregnant women is 500 µg/day [7].

## 6. Discussion

Iodine and THs play essential roles in pregnancy, ensuring proper fetal development and maternal health [3]. THs are involved in cellular development, growth, lipid and carbohydrate metabolism, cellular respiration, overall energy expenditure, growth, tissue maturation, and neurodevelopment [18]. As iodine is an essential trace element for these hormones, its adequate intake is necessary to support both the mother and the developing fetus’ brain and CNS [139,140]. During pregnancy, iodine requirements increase more than at any other stage of life, as renal excretion rises and some iodine is diverted to the placenta to pass to the fetus. Although the WHO recommends a dietary iodine intake of 150 μg per day [116], iodine requirements increase significantly to about 250 μg per day in pregnant women [8]. Iodine deficiency can have serious consequences in this context, including impaired cognitive development and reduced IQ in newborns [4,141,142], with research consistently linking iodine deficiency during pregnancy with lower IQ scores in children. Studies have shown that even mild iodine deficiency can reduce the IQ by 8 to 15 points in offspring, while chronic moderate to severe deficiency can lower the average IQ by around 13.5 points [5,143].

Iodine deficiency during pregnancy is a major public health issue, due to its irreversible effects on brain development and intelligence in children. Ensuring adequate iodine intake through diet and supplementation is crucial for optimal TH production, preventing cognitive impairments, and supporting neurodevelopment. Therefore, it is recommended that women who are pregnant, lactating, or planning pregnancy take a daily supplement of 150 μg of iodine, start supplementation three months before conception, and avoid excessive iodine doses, which can cause fetal hypothyroidism.

## 7. Conclusions and Future Perspectives

Maintaining proper iodine intake during pregnancy is crucial for healthy thyroid function, maternal health, and fetal neurodevelopment. Both severe and mild iodine deficiencies pose risks, ranging from impaired cognitive development in children to thyroid dysfunction. While physiological adaptations can partly compensate for insufficient iodine intake, they often result in thyroid overstimulation and long-term health risks for both the mother and newborn. On the other hand, excessive iodine intake can also disrupt thyroid function, underscoring the need for balanced nutritional strategies. Public health measures, such as improving access to iodized salt and appropriate supplementation, remain critical to address these challenges. Early supplementation—ideally before conception—and considering other nutritional factors (e.g., iron) is the best way to ensure maternal and fetal health. Future research should focus on developing better markers of iodine deficiency, unified guidelines, and prevention strategies to improve pregnancy outcomes and child development.

## Figures and Tables

**Figure 1 ijms-26-10247-f001:**
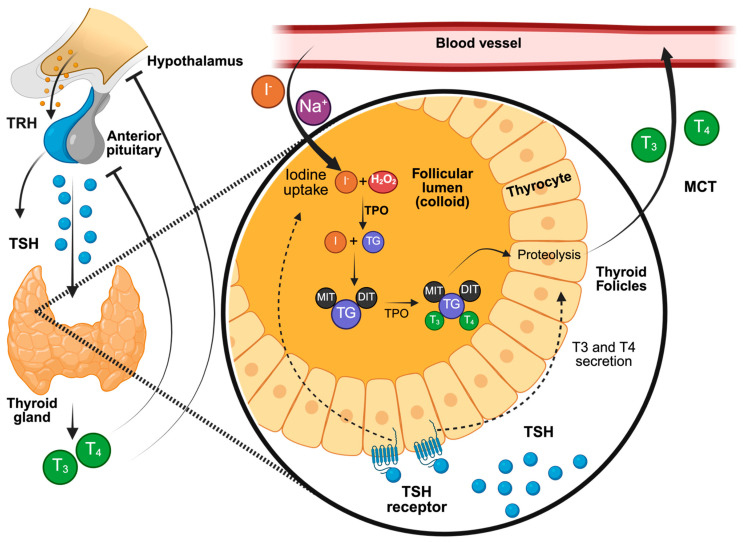
The molecular processes involved in the production and regulation of THs within the hypothalamic–pituitary–thyroid (HPT) axis. The hypothalamus releases thyrotropin-releasing hormone (TRH), which stimulates the anterior pituitary to secrete thyroid-stimulating hormone (TSH). TSH binds to its receptor on the basolateral membrane of thyrocytes, promoting iodide (I^−^) uptake via the sodium–iodide symporter (NIS). Inside the follicular lumen, thyroid peroxidase (TPO), in the presence of hydrogen peroxide (H_2_O_2_), oxidizes iodide and attaches it to tyrosyl residues on thyroglobulin (TG), forming monoiodotyrosine (MIT) and diiodotyrosine (DIT). These iodotyrosines couple to generate the thyroid hormones triiodothyronine (T3) and thyroxine (T4), which are stored in the colloid bound to TG. Following the endocytosis and proteolysis of TG, T3 and T4 are released into the bloodstream via monocarboxylate transporters (MCTs). Circulating thyroid hormones then exert negative feedback effects on both the hypothalamus and the pituitary, thereby maintaining homeostasis of the HPT axis. Created in BioRender.com (accessed on 3 February 2025).

**Figure 2 ijms-26-10247-f002:**
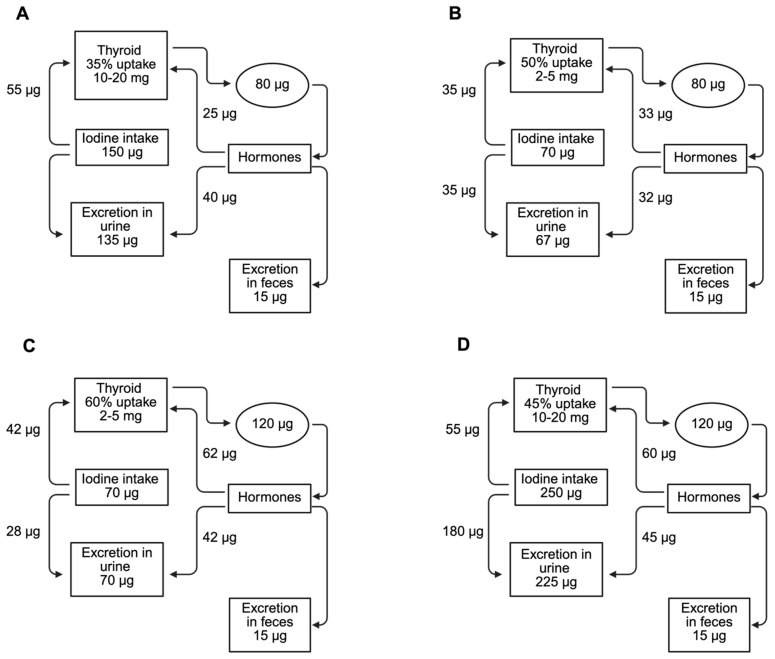
Iodide metabolism in healthy non-pregnant and pregnant women. (**A**) Non-pregnant woman with optimal iodine intake. (**B**) Non-pregnant with low iodine intake. (**C**) Pregnant women with low iodine intake. (**D**). Pregnant women with optimal iodine intake. Created in BioRender.com (accessed on 5 February 2025).

**Figure 3 ijms-26-10247-f003:**
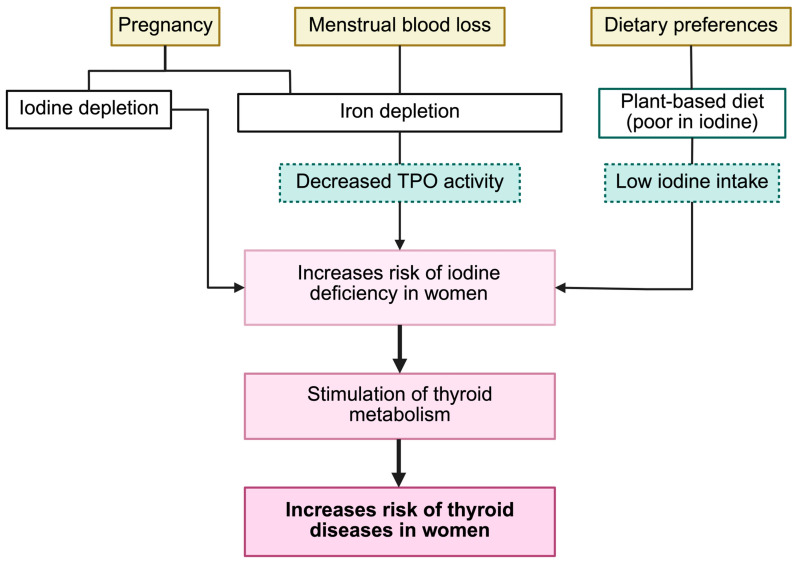
Determinants of the risk of iodine deficiency in women. Physiological (pregnancy, menstrual blood loss) and dietary (plant-based, low-iodine intake) factors promote iodine or iron depletion, reducing thyroperoxidase (TPO) activity and increasing the risk of thyroid dysfunction. Created in BioRender.com (accessed on 5 February 2025).

**Figure 4 ijms-26-10247-f004:**
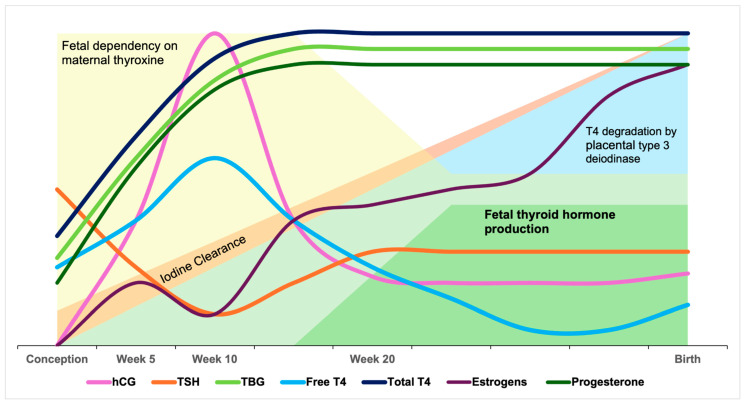
Changes in thyroid physiology during pregnancy. In early pregnancy, rising human chorionic gonadotropin (hCG) stimulates thyroxine-binding globulin (TBG) production, increases FT4 concentrations, and correspondingly suppresses TSH levels. As pregnancy progresses, placental type 3 deiodinase activity increases, leading to a decline in FT4. By around week 20, the fetal thyroid becomes sufficiently developed to produce its own hormones, reducing the dependence on maternal thyroxine. At the same time, maternal estrogen and progesterone concentrations rise to support fetal development and prepare the mother’s body for birth.

**Figure 5 ijms-26-10247-f005:**
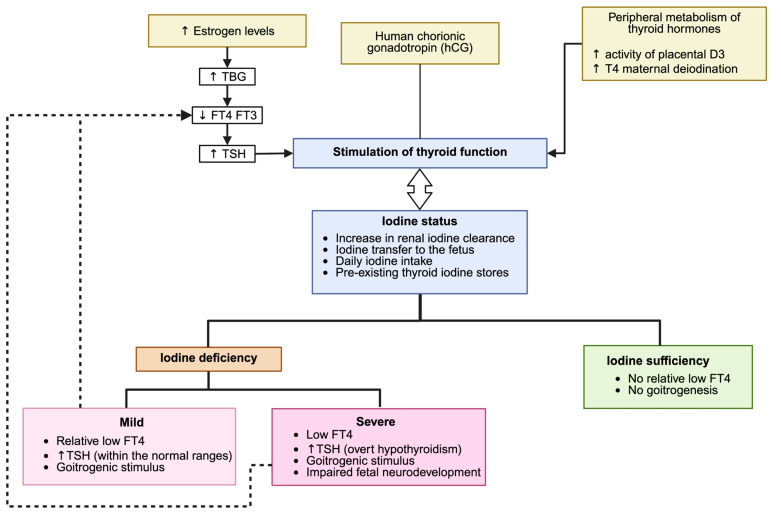
Adaptation of the thyroid function during pregnancy in iodine-sufficient and -deficient conditions. Estrogen-induced thyroxine-binding globulin (TBG) elevation, stimulation of thyroid activity by human chorionic gonadotropin (hCG), and increased maternal thyroid hormone breakdown by placental type 3 deiodinase (D3) work together to regulate the maternal thyroid physiology, with the equilibrium between iodine clearance, maternal intake, and transplacental transfer influencing the iodine status. Adequate iodine sufficiency maintains normal thyroid hormone levels and prevents goiter formation, whereas mild to severe iodine deficiency can lead to changes in FT4/TSH levels, increased goitrogenic stimulation and, in severe cases, impaired fetal neurodevelopment. Created in BioRender.com (accessed on 5 February 2025).

**Figure 6 ijms-26-10247-f006:**
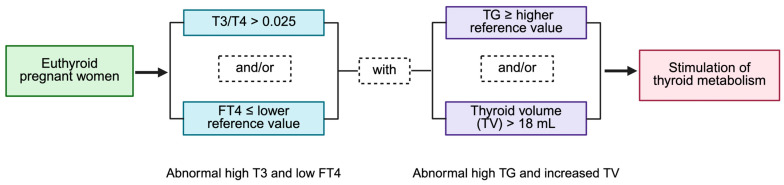
Assessing iodine adequacy in pregnant women at the individual level. Excessive thyroid stimulation in euthyroid pregnant women can be detected using routine laboratory tests and standardized criteria. A FT4 level at or below the lower reference limit, combined with a T3/T4 ratio above 0.025 and/or elevated thyroglobulin (TG ≥ higher reference value) or an increased thyroid volume (TV > 18 mL), signals a greater risk of iodine deficiency and indicates abnormal thyroid stimulation. Created in BioRender.com (accessed on 5 February 2025).

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
