# Peer review of "Critical Role of Iodine and Thyroid Hormones During Pregnancy"

_ijms, 2025, doi:10.3390/ijms262110247_

Round 1

Reviewer 1 Report

Comments and Suggestions for Authors

In present work, Reyes et al. try to review the critical role of iodine and thyroid hormones during pregnancy. This review focuses on the key mechanisms of thyroid hormone homeostasis and discusses the implications of iodine deficiency during pregnancy. However, there are some questions that should be explained.

Major concerns

1. For a review article, there are only four sections, including Introduction, Discussion, Conclusions, and References, and Main Body section should be added. Please redesign the sections.

2. In the Introduction section, the aim of this manuscript should be added.

3. Figure 1, there should be negative feedbacks. In addition, in Figure 1 legend, synthesis of thyroid hormones in thyroid follicle should be stated in detail.

4. Figure 2, please added the pregnant women with optimal iodide intake.

5. Figure 3, please added Figure legend and do not use capital letters. For the figure, cartoon images may be used with different colors.

6. Figure 4, please added the changes in progesterone and estrogen. In addition, there are significant changes in all hormones at birth.

7. Figures 5 and 6, please do not use capital letters. In addition, cartoon images may be used with different colors. Figure legend is needed.

8. English grammar and writing should be checked and revised throughout the manuscript.

Minor concerns

1. Line 2, change ‘pregnancy’ to ‘Pregnancy’.

2. Abstract section should be rewritten based on the text, which should be a refine for the text.

3. Line 43, please explain the abbreviation ‘THs,T3 and T4’. In addition, correct ‘THs ,T3 and T4(1)’.

4. Line 48, correct ‘women(3-5)’.

7. There are two ‘thyroid-stimulating hormone (TSH)’ in Lines 62 and 159.

4. Line 163, correct ‘antibodies(64-67)’ to ‘antibodies (64-67)’. Please check this throughout the manuscript.

7. There are two ‘γ-aminobutyric acid (GABA)’ in Lines 174 and 175.

6. Line 294, ‘Thyroglobulin (TG)’, but in Figure 6, ‘Tg’ is used.

7. There are four paragraphs for Conclusions section, which should be refined.

8. The formats of references are inconsistent, for example,

Some references without page number (Ref. 32, 53…).

Correct Ref. 68.

Delete ‘Thyroid: Official Journal of the American Thyroid Association’ in Ref. 76, 94, 96, and 132.

The journal names of some references are abbreviation, but others are not (Ref. 79,105, 115-117).

Ref. 115-116, do not use capital letters.

Comments on the Quality of English Language

The English could be improved to more clearly express the research.

Reviewer 2 Report

Comments and Suggestions for Authors

This is a comprehensive review article on the role of iodine and thyroid hormones during pregnancy. Gathering the most recent and relevant studies from the literature, the Authors describe the most important effects of iodine deficiency and excess on maternal and neonatal health. The Authors also explain why women are at a higher risk of exposure to iodine deficiency in comparison with men. Finally, as innovative approach, they propose to use surrogate markers of iodine status (FT4 level, T3/T4 ratio, Thyroglobulin concentration, thyroid volume) and standardized criteria to set an algorithm for selection of pregnant women who will benefit from iodine supplements.

Minor observations

At pg 3, 7, 8, 9, and 10, add a space between the legend of the figure and the text to make easier the reading.

In Fig.2, it would be very useful for the reader to see a further panel showing pregnant women with optimal iodine intake.

To avoid any confusion between individual UIC and median UIC, at line 328 change “... the UIC of school-age children is ....” with “... the median UIC of school-age children is ....”

The authors propose to use surrogate markers of iodine status and standardized criteria to select pregnant women who will benefit most from iodine supplements. This proposal is interesting and I hope that in the future the proposed algorithm (low FT4 and/or high T3/T4 ratio with high Tg and/or high thyroid volume) will be validated in large cohorts of pregnant women. However, at present, while WHO recommend iodine-containing supplements for pregnant women residing in iodine deficient areas or in areas where iodine prophylaxis programs are not consolidated yet, the ATA GL recommend that women who are planning pregnancy or currently pregnant, should supplement their diet with a daily oral supplement that contains 150 mcg of iodine (regardless their iodine nutritional status). A comment on this point would be useful.

Round 2

Reviewer 1 Report

Comments and Suggestions for Authors

Thanks for author’s responses. However, English writing should be checked and revised throughout the manuscript.

For example,

Line 48, ‘thyroid hormones (THs)’; Line 228, ‘thyroid hormone (TH)’. In addition, there are many ‘thyroid hormone’ in the text.

Line 151, delete ‘(COX-2)’. If the abbreviation is only present once, please delete and check these throughout the manuscript.

Line 161, revise ‘10-3’.

Line 344, delete ‘(MDI)’.

Line 424, delete ‘(MDI)’.

Ref. 2, 3, 4…, according MDPI style, change ‘J Clin Endocrinol Metab’ to ‘J. Clin. Endocrinol. Metab.’. Please check these throughout the Reference section.

Ref. 47, 83, and 126, delete ‘(Lausanne)’.

Ref. 71, revise ‘Journal of Perinatal Medicine’.

Ref. 73, delete ‘(Oxf)’.

Ref. 108, delete ‘(Copenh)’.

Comments on the Quality of English Language

The English could be improved to more clearly express the research.
